# Maternal Diet High in Linoleic Acid Alters Renal Branching Morphogenesis and mTOR/AKT Signalling Genes in Rat Fetal Kidneys

**DOI:** 10.3390/ijms25094688

**Published:** 2024-04-25

**Authors:** Connie McClelland, Olivia J. Holland, Nirajan Shrestha, Claire L. Jukes, Anna E. Brandon, James S. M. Cuffe, Anthony V. Perkins, Andrew J. McAinch, Deanne H. Hryciw

**Affiliations:** 1School of Pharmacy and Medical Science, Griffith University, Southport, QLD 4222, Australia; connie.mcclelland@griffithuni.edu.au (C.M.); o.holland@griffith.edu.au (O.J.H.); nirajan.shrestha@griffithuni.edu.au (N.S.); a.perkins@griffith.edu.au (A.V.P.); 2Women’s Newborn and Childrens Services, Gold Coast Hospital and Health Service, Southport, QLD 4215, Australia; 3School of Environment and Science, Griffith University, Nathan, QLD 4111, Australia; claire.jukes@griffithuni.edu.au (C.L.J.); anna.brandon@griffithuni.edu.au (A.E.B.); 4School of Biomedical Sciences, The University of Queensland, St. Lucia, QLD 4072, Australia; j.cuffe1@uq.edu.au; 5School of Health, University of Sunshine Coast, Sippy Downs, QLD 4556, Australia; 6Institute for Health and Sport, Victoria University, Melbourne, VIC 3011, Australia; 7Australian Institute for Musculoskeletal Science (AIMSS), Victoria University, St. Albans, VIC 3021, Australia; 8Griffith Institute of Drug Discovery, Griffith University, Nathan, QLD 4111, Australia

**Keywords:** linoleic acid, maternal, offspring, sex specific, kidney

## Abstract

Linoleic acid (LA), an n-6 polyunsaturated fatty acid (PUFA), is obtained from the maternal diet during pregnancy, and is essential for normal fetal growth and development. A maternal high-LA (HLA) diet alters maternal and offspring fatty acids, maternal leptin and male/female ratio at embryonic (E) day 20 (E20). We investigated the effects of an HLA diet on embryonic offspring renal branching morphogenesis, leptin signalling, megalin signalling and angiogenesis gene expression. Female Wistar Kyoto rats were fed low-LA (LLA; 1.44% energy from LA) or high-LA (HLA; 6.21% energy from LA) diets during pregnancy and gestation/lactation. Offspring were sacrificed and mRNA from kidneys was analysed by real-time PCR. Maternal HLA decreased the targets involved in branching morphogenesis *Ret* and *Gdnf* in offspring, independent of sex. Furthermore, downstream targets of megalin, namely *mTOR*, *Akt3* and *Prkab2*, were reduced in offspring from mothers consuming an HLA diet, independent of sex. There was a trend of an increase in the branching morphogenesis target *Gfra1* in females (*p* = 0.0517). These findings suggest that an HLA diet during pregnancy may lead to altered renal function in offspring. Future research should investigate the effects an HLA diet has on offspring kidney function in adolescence and adulthood.

## 1. Introduction

In Westernised societies, we are consuming more plant-based fats, with an abundance of the omega 6 (n—6) polyunsaturated fatty acid (PUFA) linoleic acid (LA; 18:2n—6; cis, cis-9,12-octadecadienoic acid) [1]. Dietary availability of LA has increased dramatically in the last 50 years [2]. LA is an essential fatty acid which is required by the body for functions such as modulating cell signalling and expression of genes, and it also has a role in inflammation [3]. LA and the omega 3 (n—3) fatty acid (FA) α—linoleic acid (ALA) are metabolised by the same enzymes [4]. A metabolite of LA is arachidonic acid (AA), which plays an integral role in the inflammatory pathway [4]. In pregnant women, the elevated intake of n—6 FAs in the Western diet is reflected in a maldistributed fatty acid profile [5]. As LA can be transported by the placenta to the foetus during development [6], a maternal elevated LA diet may impact fetal development and offsprings’ developmental outcomes.

Previous research from our group in a rodent model demonstrated that a maternal high-LA diet during pregnancy alters male/female sex ratio, circulating leptin, maternal and offspring circulating fatty acids [7] and placental FA and FA transporters [8]. Notably, there was no change in rodent offsprings’ whole organ weight [7]; however, in rodent offspring, the diet led to sex-specific cardiovascular changes [9] and hepatic changes in adolescence [10] and adulthood [11]. Importantly, leptin is critical for normal organ development [12], and adequate leptin concentrations are required for kidney development in rodents [13]. At this time, the effects of a maternal high-LA diet on offspring renal development are unknown. This is important for life-long health, as importantly, reduced nephron endowment in humans increases the risk of developing a number of diseases in adulthood [14].

Kidney development is a complex process, with a number of key targets involved in nephrogenesis. Genes regulating renal branching morphogenesis, renal growth, cellular proliferation and apoptosis are altered in developing rat kidneys due to an adverse maternal environment [14,15,16,17]. Branching morphogenesis is a process that assists in establishing the collecting ducts to the adult organ, and drives organ expansion via peripheral interactions with nephron progenitor cells [18]. Angiogenesis, which involves the development of blood vessels from the pre-existing vascular system, is crucial for normal adult kidney development. Factors which regulate development can be varied, but research has demonstrated that adipokine leptin in rodents plays a critical role in development [13]. Most of our understanding concerning leptin’s developmental influence [19] has focused on neuronal systems due to leptin’s ability to regulate appetite [20]. However, antagonism of leptin in the postnatal rodent signalling window demonstrated a reduction in the maturation of the kidneys [12]. The mechanism for this is unknown. However, leptin can bind to both the leptin receptor and megalin in the kidney [21].

Developmental programming of disease risk due to an imbalanced maternal diet during pregnancy has led to the identification of sex-specific effects in offspring outcomes [11,22,23,24,25,26]. Therefore, in this study, we aimed to investigate the effects of a maternal high-LA (HLA) diet on targets responsible for renal development in embryonic rat offspring. As developmental programming is often sex-specific [11,22,23,24,25,26], we analysed both male and female embryonic offspring independently. It was hypothesised that the kidneys of offspring from mothers consuming an HLA diet would have altered branching morphogenesis, angiogenesis and leptin and megalin signalling compared to those from offspring of mothers consuming an LLA diet, with males being more significantly affected than females.

## 2. Results

### 2.1. Effect of Maternal HLA Diet on Renal Genes Involved in Branching Morphogenesis

The organ and body weights from the E20 cohort have previously been published [7]. These data demonstrated that for both males and females, and for both left and right kidneys, a maternal HLA diet did not alter kidney weight or body weight [7]. To extend this study, we investigated renal genes involved in branching morphogenesis (Figure 1). A maternal HLA diet decreased *Ret* (Figure 1, n = 6–8, *p* = 0.0161) and *Gdnf* (Figure 1, n = 6–8, *p* = 0.0364). There was a trend towards significance for *Gfra1*, with an increase in females (Figure 1, n = 6–8, *p* = 0.0517). *Tgfb1* was unaltered in response to a maternal HLA diet (Figure 1).

### 2.2. Effect of Maternal HLA Diet on Renal Genes Involved in Leptin Signalling in Embryonic Offspring

There were no changes to *Lepr, Jak2, Stat3* or *Stat5a* in response to a maternal HLA diet (Figure 2, n = 6–8).

### 2.3. Effect of Maternal HLA Diet on Renal Genes Involved in Megalin Signalling in Embryonic Offspring

A maternal HLA diet decreased *mTOR* (Figure 3, n = 6–8, *p* = 0.0026), *Prkab2* (Figure 3, n = 6–8, *p* = 0.0477) and *Akt3* (Figure 3, n = 6–8, *p* = 0.0233). *Lrp2* (megalin), Pi3kca and *Prkaa1* were unaltered in response to a maternal HLA diet (Figure 3). Post hoc analysis indicated there was a significant decrease in *mTOR* between females from mothers consuming LLA and those from mothers consuming HLA (Figure 3).

### 2.4. Effect of Maternal HLA Diet on Angiogenesis Genes in Embryonic Kidneys

*Flt* and *Vegfa* were unaltered in response to a maternal HLA diet (Figure 4).

## 3. Discussion

In this study, we have demonstrated that a maternal HLA diet reduces the expression of genes responsible for branching morphogenesis (*Ret*, *Gdnf*) and megalin-mediated signalling (*mTOR, Akt3* and *Prkab2*). The maternal environment, particularly nutrition during pregnancy, can have significant effects on fetal growth and development [27]. Our previous research demonstrated that there are early life consequences of a maternal HLA diet [7]. Specifically, we demonstrated that leptin is reduced in mothers consuming an HLA diet. Leptin is critical for development [19]. Previous research has demonstrated that blunting the leptin postnatal peak during the first week of life in rats (corresponding to the completion of nephrogenesis) reduces the number of glomeruli [12]. However, the molecular mechanism for this was unknown. We have previously demonstrated that a maternal HLA diet reduces maternal serum leptin and alters dietary FAs [7]. To add to this, we have demonstrated that this maternal HLA diet impacts targets responsible for branching morphogenesis and mTOR/AKT signalling. This may contribute to the altered nephrogenesis observed when leptin is antagonised [12].

The change in gene expression is surprising, as in our model of a maternal HLA diet, kidney weight was unaltered at E20 [7]. Notably, kidney weight changes were also not observed at adolescence [9] or adulthood [11]. Despite this, a key finding in this study was the decreased expression of genes involved in branching morphogenesis, one of the initial stages of nephrogenesis, in the offspring of the HLA diet group. These genes, *Ret* and *Gdnf*, play an important role in the development of the kidneys [28]. *Gdnf* binds to *Ret*, activating it and allowing the process of proliferation and branching of the ureteric bud, an integral step in forming the kidney [28]. The decreased expression of *Gdnf* found in the offspring from mothers consuming an HLA diet could potentially decrease the number of nephrons formed during the nephrogenic process. Strong correlations between the early ureteric branching extent and nephron endowment have previously been found [29]. Specifically, a rodent model of food restriction in the mother resulted in decreased *Gdnf* mRNA and protein expression levels in the absence of changes to *Ret* [30]. This study is supported by a similar model where mothers consumed 50% fewer calories, which resulted in a reduction in *Gdnf* for both mRNA and protein [31]. The reduction in *Gdnf* and *Ret* has the potential to reduce the nephron number and therefore could impact kidney function later in adulthood [30]. Importantly, with the removal of *Gdnf*/*Ret* during kidney development in mice, there were significant branching abnormalities [32]. In these models of maternal food restriction, the maternal body weight and thus leptin were reduced, suggesting that leptin may regulate branching morphogenesis via a *Gdnf*/*Ret* pathway. Of interest, a previous study has shown that there are sex-specific differences in *Gdnf* in mice [33], which was not observed in our study in rats. Furthermore, in this study in mice, a maternal low-protein diet did not alter *Gdnf* mRNA [33]. Thus, the reduction in *Gdnf* in this study suggests that reduced leptin and/or altered fatty acids may impact branching morphogenesis, independently of sex.

Another novel outcome from this study was that in offspring from the maternal HLA diet group, there was decreased expression of downstream signalling targets of leptin in the megalin signalling pathway, namely *mTOR*, *Akt3* and *Prkab2*. Megalin is an endocytic receptor, highly expressed in the renal proximal tubule, and plays a vital role in the reabsorption of macromolecules filtered in the kidneys [34,35]. Megalin, in addition, mediates leptin signalling within the kidney via the *Akt/mTOR* signalling pathway to control cell metabolism, proliferation and consequently growth [36,37]. In our study, *mTOR* was significantly reduced in offspring from the HLA mothers. In a previous study, the loss of *mTOR* in nephron progenitor cells resulted in a failure to develop functional kidneys [38]. Again, surprisingly, the data in our study did not demonstrate a sex-specific effect on the *Akt/mTOR* signalling pathway.

Leptin, via the scavenger receptor megalin, also activates the *Pi3k* pathway [39]. In this study, megalin, leptin and *Pi3kca* were unaltered between treatment groups and between sexes. Similar results were observed in a uteroplacental insufficiency model of growth restriction, where megalin and its downstream signalling targets (*mTOR*, *Pi3k*, *Ampka*, *Ampkb*) at E20 were unaltered between restricted and control offspring [15]. Leptin and its downstream signalling mediators are critical in regulating cell proliferation, differentiation and growth. Therefore, the data from this study do not support the hypothesis that the leptin signalling pathway does not alter renal development in embryonic rats [15].

*mTOR* is a serine/threonine protein kinase within the *Pi3k* kinase family and serves as the subunit for two protein complexes, mTOR complex 1 (mTORC1) and mTOR complex 2 (mTORC2) [40]. mTORC1 plays a key role in the production of proteins, lipids and nucleotides, and in suppressing autophagy, all of which are essential for cell growth [40]. Furthermore, *mTOR* inhibition suppresses general mRNA translation [41]. Thus, the reduced expression of *mTOR* shown in offspring of a maternal HLA model could have potentially greater effects on protein synthesis and mRNA translation in cell growth. Additionally under nutrient-sufficient conditions, mTORC1 phosphorylates ULK1, an autophagy-initiating kinase enzyme, preventing its activation by AMP-activated protein kinase (AMPK) [39,40]. Autophagy causes degradation of cell components in order to maintain cell activity and viability under nutrient-restricted conditions [39]. With both *PRkab2* (*AMPkb*) and *mTOR* showing decreased expression in this current study, there is potential for reduced autophagy processes to occur which could possibly influence cell viability and therefore development, which warrants further investigation.

mTORC2 has a different role, controlling cell proliferation and survival via phosphorylation of several protein kinases [40]. One of the most important roles is the activation of *Akt*, a central actor in *Pi3k* signalling, via phosphorylation [40]. Upon activation of *Akt*, further phosphorylation and inhibition of key substrates occur to promote cell survival and proliferation [40]. Interestingly, in a mouse model, inhibition of mTOR with Rapamycin significantly decreased megalin, altering renal handling of proteins [42]. In this study, despite downregulation of *mTOR*, *megalin* mRNA was unaltered. Future studies should investigate the effects of the maternal HLA diet on megalin-mediated albumin endocytosis in offspring [42].

This current study showed that in fetal kidneys, leptin’s downstream signalling targets through *ObR*, including *Jak2*, *Stat3* and *Stat5a*, were not altered in response to the maternal HLA diet. Leptin is produced in adipose tissue and the placenta [19]. In rats, the leptin surge occurs from around PN5 to PN15 [43], which is the time of completion of organogenesis. Therefore, the unaltered expression of leptin signalling targets at E20 should be reinvestigated at the time of the leptin surge. In previous investigations it was found that this postnatal leptin surge that occurs was reduced in offspring in cases of maternal undernutrition [44]. Maternal undernutrition, particularly protein restriction, was also found to reduce the expression of the downstream signalling target of leptin, *Stat3* [45]. Unlike these findings, an exploration into the placenta associated with restricted mothers showed increased expression of *Jak2*, *Stat3* and *Stat5a* mRNA [46]. Since we know leptin antagonism during the leptin surge results in reduced glomerular number and size and that offspring of restricted mothers (with resulting uteroplacental insufficiency) have a reduction in expression of leptin at PN7 [12], there is a clear indication that leptin plays a role in nephron formation at PN7 [19].

## 4. Materials and Methods

### 4.1. Experimental Animal Model and Diet

Ethical approval was granted by the Griffith University Animal Ethics Committee (NSC/01/17/AEC: 26 April 2017). Wistar Kyoto rats (8 weeks of age; *n* = 8 for diet with low linoleic acid (LLA) and n = 10 for HLA diet) were purchased from the Australian Resource Centre (Kensington, WA, Australia) and housed in accordance with the Australian Code of Practice for Care and Use of Animals for Scientific Purposes, following the ARRIVE Guidelines for Reporting Animal Research [47].

Eight-week-old female Wistar Kyoto (WKY) rats were housed in individually ventilated cages under a 12 h light–dark cycle at a temperature of 20–22 °C and provided with standard food pellets during acclimatisation and tap water ad libitum throughout the study. After a week for acclimatization, female rats were randomised to consume either a control low-LA (LLA: 1.44% of energy from LA, *n* = 8) or high-LA (HLA: 6.21% of energy from LA, *n* = 8) diet for 10 weeks. The minimum requirement for LA in the rodent diet is between 1 and 1.5% [48]. The composition of the custom diet has been previously reported [7]. These diets were isocaloric and matched for n-3 PUFA and total fat content. Pregnant females were euthanised at E20 via terminal anaesthetisation with intraperitoneal injections of sodium pentobarbital. The foetuses were euthanised by decapitation and fetal kidneys were harvested and stored at −80 °C following snap freezing in liquid nitrogen.

### 4.2. Quantitative Real-Time Polymerase Chain Reaction (qPCR)

Sex determination was performed as described previously [7]. Total RNA was extracted from kidney tissue using the RNeasy Mini kit (Qiagen, Chadstone, VIC, Australia) following the manufacturer’s guidelines. The quantification and evaluation of purity of RNA samples were conducted using a NanoDrop 1000 spectrophotometer (Thermo Fisher Scientific, Waltham, MA, USA). Reverse transcription of RNA to synthesize complementary DNA was performed using the iScript gDNA clear cDNA synthesis kit (BioRad, Hercules, CA, USA) following the manufacturer’s guidelines. Quantitative PCR was performed using QuantiNova SYBR^®^ green master mix (Qiagen) following manufacturer’s guidelines, in line with the Minimum Information for Publication of Quantitative Real-Time PCR Experiments (MIQE) guidelines [49]. PCR initial heat activation was run for 2 min at 95 °C, then qPCR reactions were run for 40 cycles of 95 °C for 5 s (denaturation) and 60 °C for 10 s (combined annealing/extension) using StepOne^TM^ real-time PCR systems (Applied Biosystems, Waltham, MA, USA). Gene expression of targets (Appendix A) was quantified using the 2^−ΔΔ*C*q^ method normalised to the geometric mean of β-actin and β-2 microglobulin as reference genes. These reference genes were stable across the treatment groups.

### 4.3. Statistical Analysis

All data were analysed using GraphPad Prism 8.3.1. One male and one female offspring from each litter were analysed. n values represent individual offspring from separate litters. Data were analysed separately for males and females, with both sexes analysed by two-way ANOVA with maternal and postnatal diet as the factors. Specific comparisons were made using Tukey’s post hoc test. Data are presented as mean ± standard error of the mean (SEM). *p*-values < 0.05 were considered evidence of significant differences.

## 5. Conclusions

In conclusion, we have demonstrated that a maternal HLA diet in rodents alters genes responsible for branching morphogenesis and *mTOR/AKT* signalling. As these changes occurred independently of sex, this suggests that the reduction in maternal leptin and/or altered fatty acids significantly impacted embryonic nephrogenesis. Further studies should be conducted to identify if the maternal HLA diet alters adolescent or adult offspring renal function, in addition to offspring renal histological changes associated with a maternal HLA diet during pregnancy.

## Figures and Tables

**Figure 1 ijms-25-04688-f001:**
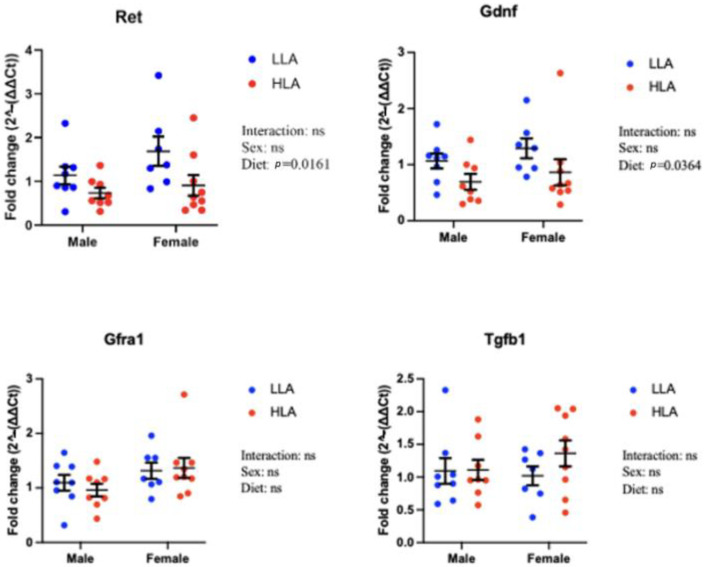
Effect of maternal diet high in linoleic acid on branching morphogenesis genes in the kidneys of embryonic offspring. Data are presented as mean ± standard error of the mean (SEM). Two-way ANOVA was performed for statistical analysis with maternal diet and sex as two factors. n = 6–8. LLA: low linoleic acid; HLA: high linoleic acid. ns = not significant.

**Figure 2 ijms-25-04688-f002:**
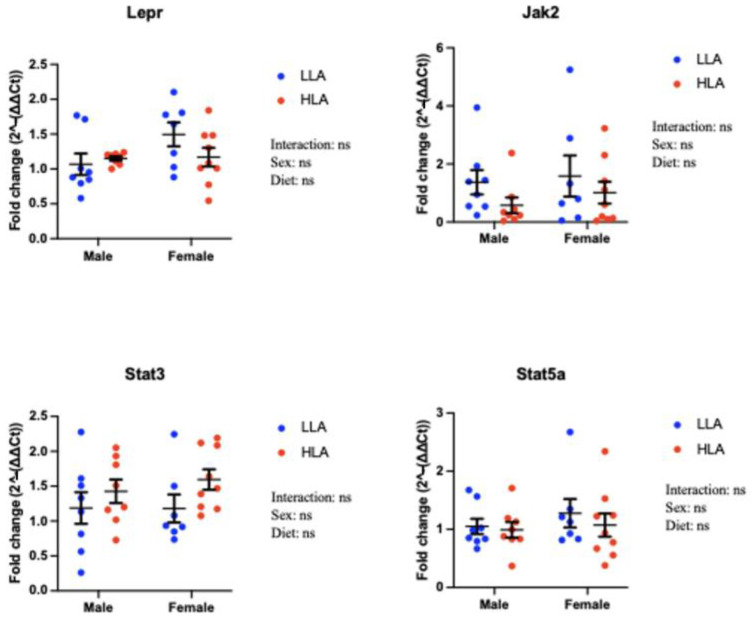
Effect of maternal diet high in linoleic acid on leptin signalling genes in the kidneys of embryonic offspring. Data are presented as mean ± standard error of the mean (SEM). Two-way ANOVA was performed for statistical analysis with maternal diet and sex as two factors. n = 6–8. LLA: low linoleic acid; HLA: high linoleic acid. ns = not significant.

**Figure 3 ijms-25-04688-f003:**
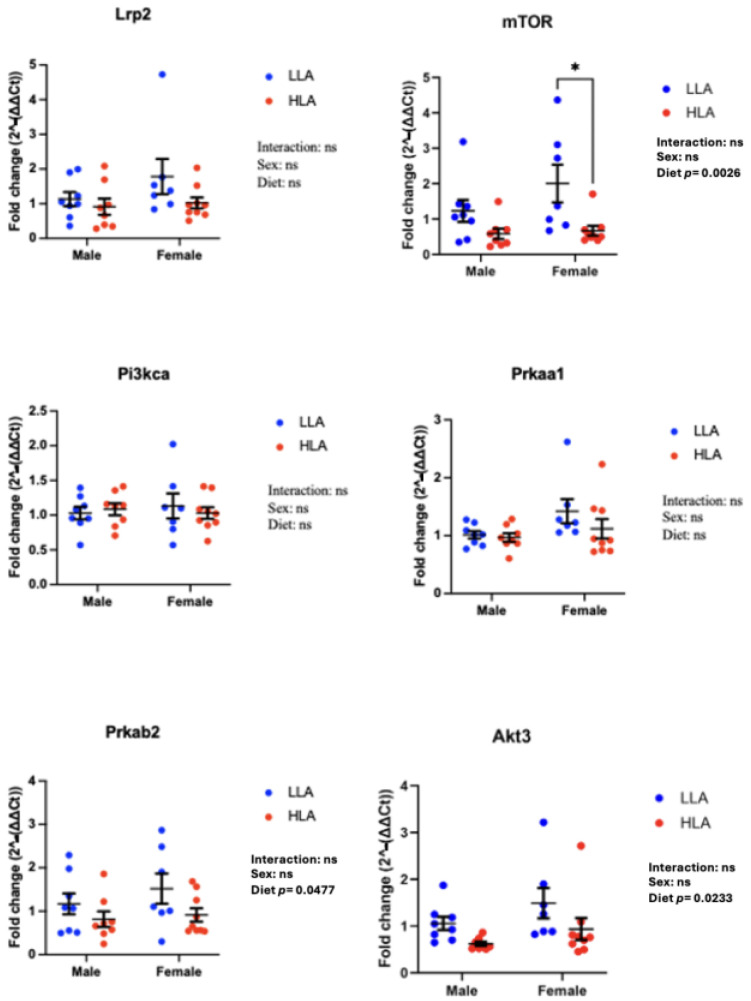
Effect of maternal diet high in linoleic acid on megalin signalling genes in the kidneys of embryonic offspring. Data are presented as mean ± standard error of the mean (SEM). Two-way ANOVA was performed for statistical analysis with maternal diet and sex as two factors. n = 6–8. LLA: low linoleic acid; HLA: high linoleic acid. Where post hoc analysis identified a difference, differences across the groups are denoted by an asterisk (* *p* < 0.05). ns = not significant.

**Figure 4 ijms-25-04688-f004:**
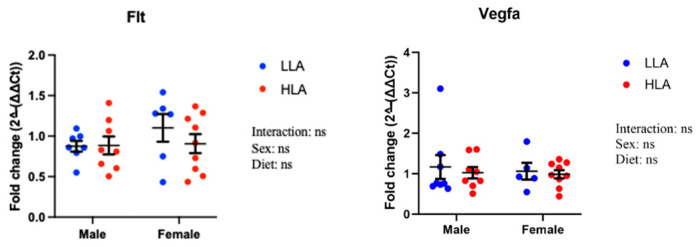
Effect of maternal diet high in linoleic acid on angiogenesis genes in the kidneys of embryonic offspring. Data are presented as mean ± standard error of the mean (SEM). Two-way ANOVA was performed for statistical analysis with maternal diet and postnatal diet as two factors. n = 6–8. LLA: low linoleic acid; HLA: high linoleic acid. ns = not significant.

## Data Availability

Data are contained within the article and Appendix A.

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
