# Peer review of "Maternal Diet High in Linoleic Acid Alters Renal Branching Morphogenesis and mTOR/AKT Signalling Genes in Rat Fetal Kidneys"

_ijms, 2024, doi:10.3390/ijms25094688_

Round 1
Reviewer 1 Report
Comments and Suggestions for Authors
The article entitled “Maternal diet high in linoleic acid alters renal branching morphogenesis and mTOR/AKT signaling genes in rat fetal kidneys.” (ijms-2962070) has been presented to the section “Molecular Biology”, the thematic of this manuscript is agree with the topic of this section.
This study investigates the effects of a HLA diet on renal branching morphogenesis, leptin signaling, megalin signaling, and angiogenesis gene expression in rat embryonic offspring. The findings suggest that exposure to an HLA diet during pregnancy may lead to altered renal function in offspring.
Introduction: The introduction integrates relevant literature on the topic, clearly articulating the significance of the research to be conducted. However, it is advisable to conclude the introduction by succinctly stating the research hypothesis and objectives, while also distinguishing between studies conducted in humans and experimental animals. Additionally, some of the referenced literature in the current manuscript may not directly correspond to the study's focus.
Results: The results are presented in structured sections, enhancing comprehension of the topic. The figures provided are informative and aid in understanding the obtained results. However, including quantitative data in a table format would further enhance clarity, allowing for easier interpretation of statistically significant differences.
Discussion: In the initial paragraph of the discussion, it is customary to highlight the achievements of the presented work rather than those of previously published articles. Therefore, it is recommended to revise this section accordingly.
Materials and Methods: This section outlines the ethical approval process and describes the methodology employed in the study.
Conclusion: It is pertinent to mention that the evaluation was conducted in rats, implying the necessity for future studies to assess this scenario in the human species as well.
Reviewer 2 Report
Comments and Suggestions for Authors
1. A table to note the reference clones from Genbank (access numbers) should be presented in supplementary results. .
2. A table to show the basic information such as body weight change of the females, fetus weight and relative kidney weight should be presented.
3. Western blot or immunohistochemistry studies to confirm the qRT-PCR results in Figure 1 and 3 are required.
4. Also, H&E staining studies to show the renal branching morphogenesis by HLA are required.
5. Immunostaining studies to show the structures and the major sites of embryonic kidney by HLA effects are required.
6. In vitro studies using an optimal cell line to confirm the results of Figure 1 and 3 are suggested.
Round 2
Reviewer 2 Report
Comments and Suggestions for Authors
The present results are suggested to be combined with future studies regarding to the histological changes, for a more comprehensive study.
Comments on the Quality of English LanguageMinor editing of English language required
Author Response
Dear Reviewer
We understand the request to add immunohistochemical analysis, and appreciate their enthusiasm for the field and advancing the knowledge on this topic.
However as we indicated, we did not collect kidneys at the time of sacrifice that were preserved for immunohistochemical analysis. To repeat the experiments to generate new kidneys for this type of analysis would be beyond the scope of the study, and not able to be undertaken considering the considerable time and budget (which we do not have) to be able to repeat these experiments. We have acknowledged the limitations by suggesting future work on this topic.
Further, in alignment with the 3Rs associated with animal research https://www.nhmrc.gov.au/research-policy/ethics/animal-ethics/3rs, we feel that at this time these additional experiments are not needed.
Sincerely
A/P Hryciw
Round 3
Reviewer 2 Report
Comments and Suggestions for Authors
In the agrument of biological science, validations from multiple aspects are required. Only qRT-PCR results are not convincing.
Comments on the Quality of English LanguageMinor editing of English language required